# Single-Line Bidirectional Optical Add/Drop Multiplexer for Ring Topology Optical Fiber Networks

**DOI:** 10.3390/s21082641

**Published:** 2021-04-09

**Authors:** Chung-Yi Li, Ching-Hung Chang, Zih-Guei Lin

**Affiliations:** 1Department of Communication Engineering, National Taipei University, New Taipei City 23741, Taiwan; cyli@gm.ntpu.edu.tw; 2Department of Electrical Engineering, National Chiayi University, Chiayi 60004, Taiwan; tklingary@gmail.com

**Keywords:** single-line bidirectional optical add/drop multiplexer, optical fiber transport system, optical circulator, fiber link failure

## Abstract

A new type of passive single-line bidirectional optical add/drop multiplexer (SBOADM) is proposed and experimentally demonstrated. When the proposed SBOADM is placed as a node of a ring topology optical fiber network, the special routing function of the SBOADM can always drop down the desired downstream signals whether the signals are injected into the SBOADM in either the clockwise (CW) or counterclockwise (CCW) direction and can upload and send back the upstream signals via the reversed optical pathway of the downstream signals. Once fiber link failure occurs in the optical network, the blocked network connections can be recovered immediately by sending out the downstream signals in both the CW and CCW directions of the fiber ring. As in all passive devices, the SBOADM needs no power supply or complicated network management to achieve the bidirectional function. Thus, the proposed device is an optimal solution to enhance the stability and reliability of rapidly developed optical fiber networks.

## 1. Introduction

With the rapid development of science and technology, a variety of network applications, such as 5G communications, have emerged. The number of base stations (BSs) must be dramatically boosted to support the increased network capacity in the same network coverage range, and the stability of network transmission and reliability need to be enhanced. Radio over fiber (RoF) transport systems are suggested as an ideal broadband wireless network solution to support such a scenario because of their superior characteristics, such as high-frequency bandwidth, low attenuation and resistance to electromagnetic interference [1,2,3]. Therefore, numerous investigations on fiber 5G convergent systems [4,5,6] have been developed to overcome the limitations of shortening the transmission distance due to the increase in data transmission bandwidth in wireless communication [7].

The traditional fiber-optic network architecture is mainly designed in tree topology because of cost considerations. However, such a topology has a poor performance in the reliability and the flexibility of adding or removing network nodes. When using the current network architecture whilst considering the cost of equipment build-up, some researchers have suggested that a mixed tree and ring network architecture may be a reasonable solution [8]. In a ring-shaped architecture, considering scalability and system reliability is necessary; thus, the optical add/drop multiplexer (OADM) is a good choice [9,10,11]. However, OADM-based optical transport systems only support the single transmission direction, and the traditional bidirectional OADM-based optical network requires the additional deployment of spare optical fiber loops in the transport system to ensure network connection during fiber link failure [12,13,14]. Such devices cannot maximize the potential of utilizing bidirectional transmissions to overcome network traffic disconnections during fiber link failure in a ring topology network. Therefore, other researchers have proposed that reconfigurable OADM (ROADM), which can meet the needs of network wavelength allocation, can achieve the characteristics of capturing arbitrary wavelengths. However, building ROADM in the access network is not suitable due to its high cost and complex structure [15,16]. Several types of single-line bidirectional add/drop multiplexers (SBOADMs) [17,18,19,20,21,22,23,24,25], which can operate bidirectional transmissions on a single fiber loop, are later developed. The overall network construction cost can be reduced by removing the spare optical fiber loop. Nevertheless, such SBOADMs need the assistance of optical switches, optical multiplexers or de-multiplexers to achieve bidirectional function in a ring topology network. When SBOADMs are employed to deploy RoF transport systems, the overall cost and insertion loss caused by such SBOADMs are unignorable. A complicated strategy is required to manage the status of various optical switches inside such SBOADMs and reset the network route pathway dynamically during fiber link failure.

The SBOADM, as a network node placed in ring topology-based optical fiber transport systems, becomes appropriate if its architecture is further improved. A new form of SBOADM is proposed on the basis of four 4-port optical circulators (OCs) and two fiber Bragg gratings (FBGs) to achieve the aforementioned target. The feature of the novel self-developed SBOADM is that targeted optical signals can always be dropped down and the added upstream signal can be sent out along the reverse transmission direction of the dropped signal whether these signals are injected in either the clockwise (CW) or counterclockwise (CCW) direction. Thus, optical signals generated from the central office (CO) can be transmitted to each SBOADM node in either CW or CCW directions of the ring structure network, and the upstream signals added by SBOADMs can be sent back to the CO in the reverse optical pathways to eliminate the impact of fiber link failure. Complicated network management and additional power are unnecessary because the overall architecture of the SBOADM comprises passive components.

## 2. Experimental Setup

The schematic of the proposed SBOADM architecture is shown in Figure 1. This architecture uses only two types of common passive components: four 4-port OCs and two FBGs with 99% reflectivity (the parameters are shown in Table 1 in detail). The red auxiliary line path (dropped signal) in Figure 1a shows that the downlink optical signal will be routed to FBG1 through OC1 (entered from port 2 and outputted from port 3) when the downlink optical signal is inputted to the I/O_P1 port of this new type SBOADM. Subsequently, the optical signal located at the FBG1 Bragg wavelength will be reflected back to OC1 (entered from port 3 and outputted from port 4) and OC2 (enter from port 1 and output from port 2) and then routed to the Add/Drop_1 port. In addition, the other downlink optical signals outside the FBG1 Bragg wavelength range will be guided through FBG1 and OC3 (enters from port 2 and outputted from port 3) and outputted to the I/O_P2 port, as indicated by the blue auxiliary line path (Passed signal) in Figure 1a. Simultaneously, when the uplink optical signal enters from the Add/Drop_1 port, this signal will be guided through OC2 (enters from port 2 and outputted from port 3) and then routed to FBG2. When the uplink optical carrier wavelength is located at the FBG2 Bragg wavelength range, the uplink optical signal will be reflected back to OC2 by FBG2 and outputted by OC2 (enters from port 3 and outputted from port 4) and OC1 (enters from port 1 and outputted from port 2) to the I/O_P1 port, as shown in the green auxiliary line path (added signal) in Figure 1a.

For optical signals inputted from the I/O_P2 port of the proposed SBOADM, non-target optical signals, as shown in the blue auxiliary line path (Passed signal) in Figure 1b, will pass through OC3 (enters from port 3 and outputted from port 4), OC4 (enters from port 1 and outputted from port 2), FBG2, OC2 (enters from port 3 and outputted from port 4) and OC1 (enters from port 1 and outputted from port 2) before being routed to the I/O_P1 port. Optical signals at target wavelengths inserted from the I/O_P2 port also pass through OC3 (enters from port 3 and outputted from port 4) and OC4 (enters from port 1 and outputted from port 2) but are reflected back to OC4 (enters from port 2 and outputted from port 3) by FBG2 and outputted to the Add/Drop_2 port, as shown in the red auxiliary line path (dropped signal) of Figure 1b. When the optical signal is inputted through the Add/Drop_2 port, this signal will be guided through OC4 (enters from port 3 and outputted from port 4) and OC3 (enters from port 1 and outputted from port 2) and then reflected back to OC3 (enters from port 2 and outputted from port 3) via FBG1 before being outputted to the I/O_P2 port, as shown in the green auxiliary line path (added signal) in Figure 1b.

Figure 2a shows that five downstream optical carriers are fed into the new type SBOADM via I/O_P1 and I/O_P2 ports to evaluate the drop/add function of the proposal, and one upstream optical carrier is added in the SBOADM via Add/Drop_1 and Add/Drop_2 ports. Each downstream optical carrier has the same power of 0.06 dBm but there are different wavelengths whose center wavelengths are 1544.540, 1546.540, 1548.540, 1550.540 and 1552.540 nm. Each optical carrier is modulated with 1.2 Gbps/10 GHz radio-frequency (RF) signal via a Mach–Zender modulator. The third downstream optical carrier is also reused as an upstream optical carrier (as shown in Figure 2b) to simplify the experiment, and the Bragg wavelengths of the FBG1 and FBG2 are set at the same range of 1548.540 nm.

## 3. Experimental Results and Discussion

When five downlink optical signals inputted from the I/O_P1 port of the proposed SBOADM and one uplink optical signal inputted from the Add/Drop_1 port, the third downlink optical signal will be intercepted in accordance with the FBG1 Bragg wavelength and dropped to the Add/Drop_1 port. Similarly, the other downlink optical signals will pass through the SBOADM and outputted from the I/O_P2 port. The optical spectra measured at the Add/Drop_1 and I/O_P2 ports are shown in Figure 3a,b, respectively. For the passed signals observed at the I/O_P2 port, the insertion loss caused by the proposed SBOADM is roughly 1.66 dB (0.06 − (−1.6) = 1.66) and 21.52 dB (0.06 − (−1.6) + 19.86 = 21.52) for the un-dropped and dropped optical signals, respectively, as shown in Figure 3b. The power variation between the two types of signals is 19.86 dB, which is mainly caused by the reflection characteristic of the FBG1. Similarly, the peak power level of the target signal is −2.76 dBm as shown in Figure 3a to observe the optical signals dropped to the Add/Drop_1 port. Comparing the dropped signal with the original signals shown in Figure 2a, the target and non-target signals suffer roughly 2.82 dB (0.06 − (−2.76) = 2.82) and 39.95 dB (2.82 + 37.13 = 39.95) insertion loss, respectively, resulting in roughly 37.13 dB power variation amongst these signals. The large power variation value will significantly reduce the unwanted interference amongst the target and non-target signals. The electrical spectra of the detected downlink light signal, its frequency down-converted signal and the detected eye diagram are shown in Figure 4a,b. The transmitted RF signal is observed in the 10 GHz area and the signal quality is ensured by the open and clear eye diagram. By contrast, when the upstream optical signal is fed to the Add/Drop_1 port, the optical spectrum measured at the I/O_P1 port is shown in Figure 5a, and its electrical spectrum and eye diagram after PD detection and frequency down-conversion are shown in Figure 5b. The light power of the measured optical signal is −2.98 dBm. Thus, when an optical signal is added into a network via the proposed SBOADM, this signal will suffer roughly 3.04 dB (0.06 − (−2.98) = 3.04) insertion loss, which is roughly equal to a 2 × 1 optical coupler. Similar to the dropped signal, the transmitted RF signal is measured as presented in the electrical spectrum, and the obtained eye diagram is open and clear.

The five downstream signals and one upstream signal are respectively fed into the SBOADM from the I/O_P2 port and add/Drop_2 port to simulate the transmission of the optical signals in the CCW direction of a ring topology network. The optical spectra of the dropped and passed signals measured at the Add/Drop_2 and I/O_P1 ports are shown in Figure 6a,b, respectively. The power level of the non-dropped and dropped wavelengths are roughly −3.4 and −21.62 dBm to observe the passed signals shown in Figure 7b, resulting in 18.22 dB power variation amongst these wavelengths. Compared with the original signals shown in Figure 2a, the overall insertion loss for the non-dropped and dropped signals are 3.46 dB (0.06 − (−3.4) = 3.46) and 21.68 dB (0.06 − (−21.62) = 21.68), respectively. The power level of the target signal is reduced to −2.64 dBm whilst the values of the non-target signals are reduced to around −40 dBm to observe the optical signals at the Add/Drop_2 port. Roughly 37.6 dB power deviation is present amongst these signals. This power deviation is mainly caused by the frequency response of the FBG2 inside the SBOADM. The carried RF signal can be properly converted back to the electrical domain because the power of targeted signals is substantially larger than that in non-targeted signals. The observed electrical spectra of the received RF and frequency down-converted signals and the eye diagram are shown in Figure 7a,b, respectively. The obtained eye diagram is as clear as that shown in Figure 4b. This finding proves that the proposed SBOADM can always drop down target optical signals either via the I/O_P1 port or I/O_P2 port. Similar to previous scenarios, the upstream optical signal will be routed to the I/O_P2 port when it is uploaded from the Add/Drop_2 port. The optical spectrum measured at the I/O_P2 port is shown in Figure 8a, and the electrical spectrum and eye diagram of the obtained uplink signal after PD detection and frequency down-conversion are shown in Figure 8b. Although the added optical signal suffers 3.62 dB (0.06 − (−3.56) = 3.62) insertion loss compared with the original input optical signal, the modulated upstream RF signal is also properly received and measured.

The eye diagrams were measured over a period of time. The clear and open eyes will be great evidences to ensure the quality and stability of the SBOADM to be placed in optical fiber transport systems or fiber sensor networks. When the proposed SBOADM is placed in a ring topology network, no complicate management or power supply is required to achieve the add/drop function. The 1.2 Gbps/10 GHz signal is used mainly to verify the transmission feasibility of the proposed architecture. The overall Bragg wavelength range of FBGs within the SBOADM can be extended by a wider–bandwidth FBG or cascading multiple FBGs. As a result, different modulation format, such as 400 Gbps PAM or OFDM signals, can also been employed in the SBOADM-based optical fiber network.

Although the FBGs in 1550 nm may suffer about 10 pm/℃ Bragg wavelength shifting, the effect can be eliminated by adding an appropriate thermal control scheme or simply by extending the equivalent Bragg wavelength range by cascading two or more different FBGs. For example, embedding an additional 0.4 nm Bragg wavelength range will eliminate the effects of 40 °C temperature drifting. Besides, if there is more than one stream need to be added/dropped, the system manager can simply replace the FBGs inside the SBOADM or cascade extra FBGs with the original ones to modify the SBOADM characteristics. Once the reflection center wavelengths of FBGs are set apart from each other, the crosstalk or interference between different wavelength channels can be minimized.

## 4. Conclusions

A new type of SBOADM is proposed on the basis of four 4-port OCs and two FBGs to support the stability and reliability of ring topology-based optical fiber transport systems. When the proposed SBOADM is placed at a ring topology network, CO can transmit downstream optical signals to each SBOADM via either the CW or CCW direction of the ring structure network. The SBOADM can always drop down desired optical signals and allow the passage of non-targeted optical signals. Similarly, the upstream signals added in the ring-based network via the SBOADM node can be sent back to the CO via the reversed downstream optical pathways. Consequently, the ring topology-based optical fiber transport system can automatically eliminate the impact of any fiber link failure when the downstream signals are transmitted in the CW and CCW directions of the fiber ring. No power supply or complicated network management is required. The experimental results prove that the proposed SBOADM can properly accomplish the bidirectional routing functions without optical switches, optical multiplexers or de-multiplexers. The overall insertion loss due to the proposed SBOADM is roughly equal to a 1 × 2 optical splitter. The proposed system is a suitable device to enhance the stability and reliability of rapidly developed optical fiber networks.

## Figures and Tables

**Figure 1 sensors-21-02641-f001:**
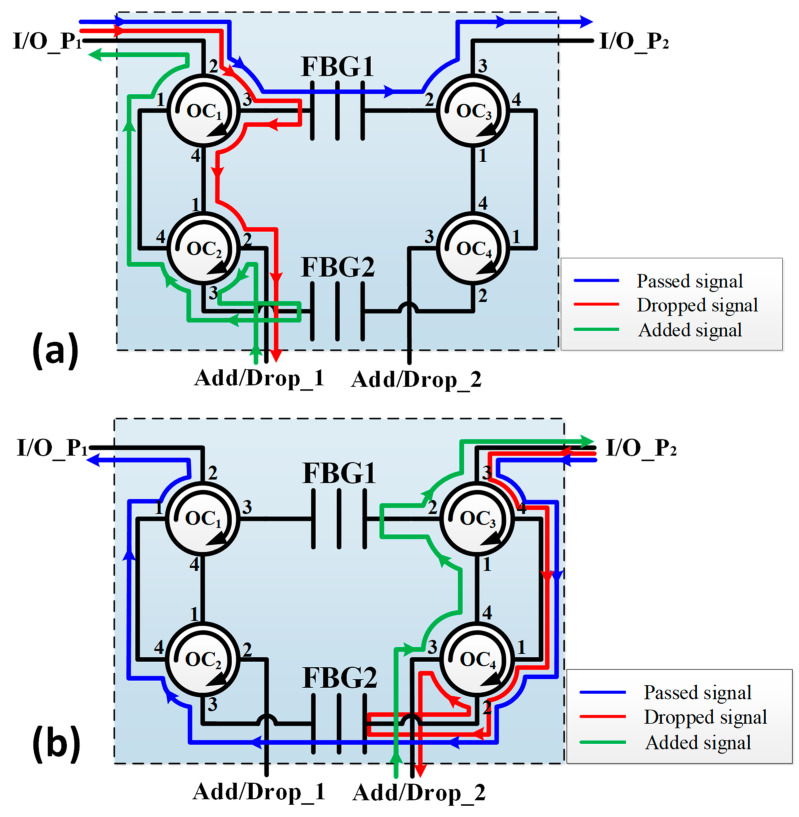
Schematic of the proposed single-line bidirectional optical add/drop multiplexer (SBOADM) in the following cases: (**a**) optical signals inserted from the left-hand side and (**b**) optical signals inserted from the right-hand side.

**Figure 2 sensors-21-02641-f002:**
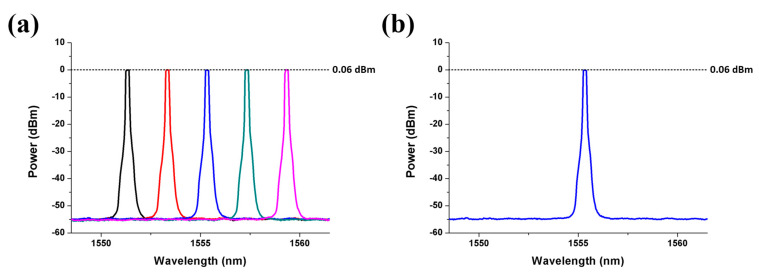
Spectral diagram of (**a**) five downlink 1.2 Gbps/10 GHz optical signal and (**b**) one uplink optical signal.

**Figure 3 sensors-21-02641-f003:**
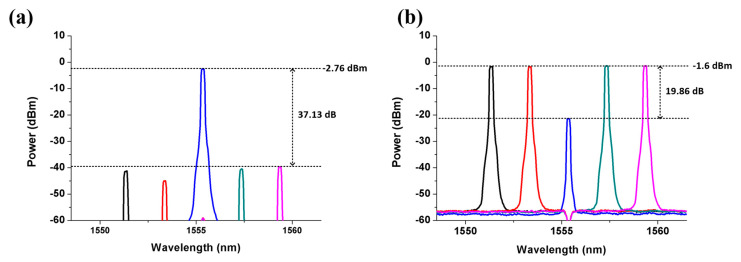
Optical spectra measured at the (**a**) Add/Drop_1 port and (**b**) I/O_P2 port when five downlink optical signals are fed in the I/O_P1 port.

**Figure 4 sensors-21-02641-f004:**
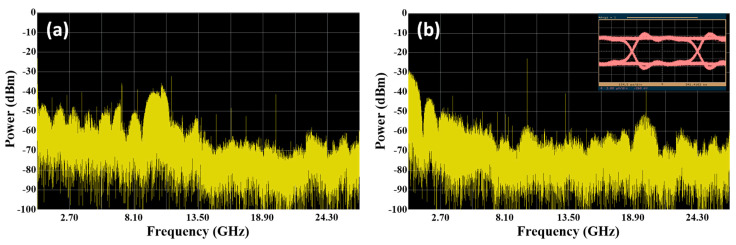
(**a**) Electrical spectrum of the detected downlink light signal measured at the Add/Drop_1 port and (**b**) its frequency down-converted signal and eye diagram.

**Figure 5 sensors-21-02641-f005:**
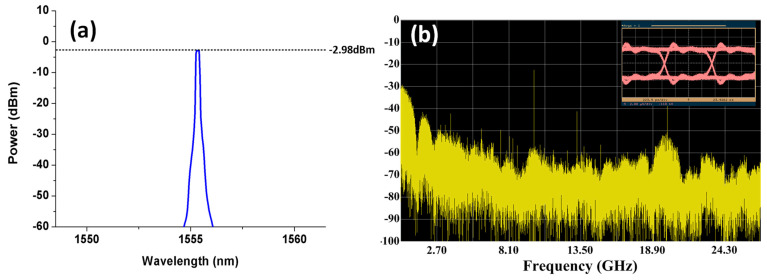
(**a**) Optical spectrum of the uplink signal measured at the I/O_P1 port and (**b**) its electrical spectrum and eye diagram after PD detection and frequency down-conversion.

**Figure 6 sensors-21-02641-f006:**
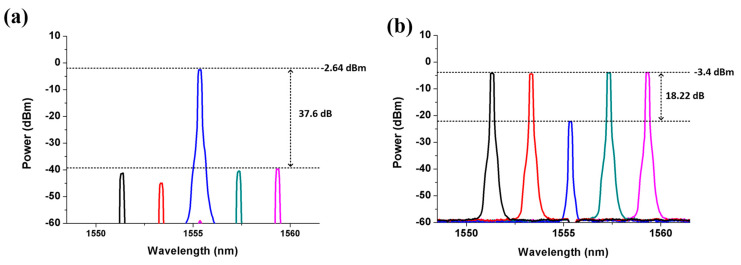
Optical spectra measured at the (**a**) Add/Drop_2 port and (**b**) I/O_P1 port when five downlink optical signals are fed into the I/O_P2 port.

**Figure 7 sensors-21-02641-f007:**
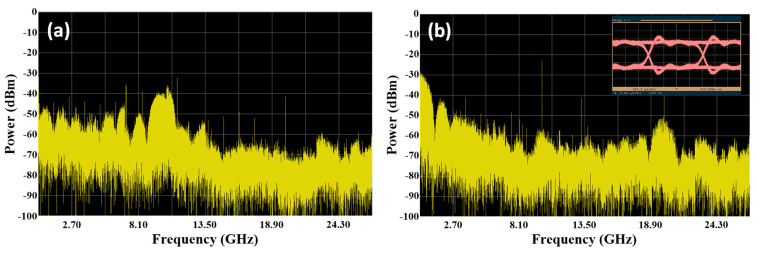
(**a**) Electrical spectrum of detected downlink light signal measured at the Add/Drop_2 port and (**b**) its frequency down-converted signal and eye diagram.

**Figure 8 sensors-21-02641-f008:**
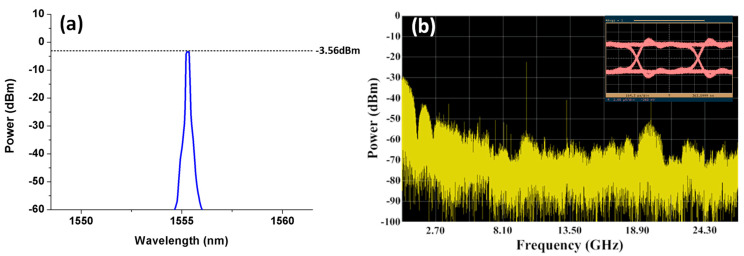
(**a**) Optical spectrum of the uplink signal measured at the I/O_P2 port and (**b**) its electrical spectrum and eye diagram after PD detection and frequency down-conversion.

**Table 1 sensors-21-02641-t001:** System parameters.

Component	Optical Circulators (OC)	Fibre Bragg Gratings (FBG)
Data Sheet	Insertion Loss (dB): <1.3 Isolation (dB): >35	Reflectivity (%): 99.25 Reflection Bandwidth (GHz): 25
System Dimensions (mm)	150 × 90 × 60 (L × W × H)	

## Data Availability

Not applicable.

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
