# Peer review of "Single-Line Bidirectional Optical Add/Drop Multiplexer for Ring Topology Optical Fiber Networks"

_sensors, 2021, doi:10.3390/s21082641_

Round 1
Reviewer 1 Report
The authors proposed a type of single-line bidirectional optical add/drop multiplexer (SBOADM) to support the stability and reliability of ring topology optical fibre transport systems. Only two types of commercial passive devices are used. Theoretical analysis and experimental verifications were made, which demonstrate the ability of the system. This paper may be accepted for publication if the following issues are well addressed.
- It would be better for the authors to add some relative SBOADM schemes using FBGs and/or OCs.
- I suggest the authors show the BER curve results of the detected downlink or uplink signals. Such curves can give a clearer explanation of the quality and stability of the signals.
- It seems the signals in Figure 4 (a) and Figure 7 (a) have high noise level in the low frequency range. Why does this happen?
- Some details about the FBGs should be given and what’s the optimal reflection characteristic of the FBGs?
- What’s the influence of temperature variation in the proposed scheme? As we know the wavelengths of the FBGs are temperature sensitive.
- How about when more than one streams need to be added/dropped?
Author Response
Manuscript ID: sensors-1137692
Title: Single-Line Bidirectional Optical Add/Drop Multiplexer for Ring Topology Optical Fibre Networks
Author: Chung-Yi Li, Ching-Hung Chang *, Zih-Guei Lin
The Reply to Reviewer 1
We thank the reviewer for the comments and valuable suggestions. The manuscript has been revised accordingly. All modifications are indicated with “underline” for reviewing convenience.
Comments to the Author
The authors proposed a type of single-line bidirectional optical add/drop multiplexer (SBOADM) to support the stability and reliability of ring topology optical fibre transport systems. Only two types of commercial passive devices are used. Theoretical analysis and experimental verifications were made, which demonstrate the ability of the system. This paper may be accepted for publication if the following issues are well addressed.
- It would be better for the authors to add some relative SBOADM schemes using FBGs and/or OCs.
Response: The second paragraph of the Introduction section is modified accordingly and additional references have been added.
The traditional fiber-optic network architecture is mainly designed in tree topology because of cost consideration. However, such a topology has poor performance in the reliability and the flexibility of adding or removing network nodes. When using the current network architecture whilst considering the cost of equipment build-up, some researchers have suggested that a mixed tree and ring network architecture may be a reasonable solution [8]. In a ring-shaped architecture, considering scalability and system reliability is necessary; thus, the optical add/drop multiplexer (OADM) is a good choice [9-11]. However, OADM-based optical transport systems only support the single transmission direction, and the traditional bidirectional OADM-based optical network requires additional deployment of spare optical fiber loops in the transport system to ensure network connection during fiber link failure [12-14]. Such devices cannot maximize the potential of utilizing bidirectional transmissions to overcome network traffic disconnections during fiber link failure in a ring topology network. Therefore, other researchers have proposed that reconfigurable OADM (ROADM), which can meet the needs of network wavelength allocation, can achieve the characteristics of capturing arbitrary wavelengths. However, building ROADM in the access network is not suitable due to its high cost and complex structure [15-16]. Several types of single-line bidirectional add/drop multiplexers (SBOADMs) [17-25], which can operate bidirectional transmissions on a single fiber loop, are later developed. The overall network construction cost can be reduced by removing the spare optical fiber loop. Nevertheless, such SBOADMs need the assistance of optical switches, optical multiplexers or de-multiplexers to achieve bidirectional function in a ring topology network. When SBOADMs are employed to deploy RoF transport systems, the overall cost and insertion loss caused by such SBOADMs are unignorably. A complicated strategy is required to manage the status of various optical switches inside such SBOADMs and reset the network route pathway dynamically during fiber link failure.
The SBOADM, as a network node placed in ring topology-based optical fiber transport systems, becomes appropriate if its architecture is further improved. A new form of SBOADM is proposed on the basis of four 4-port optical circulators (OCs) and two fiber Bragg gratings (FBGs) to achieve the aforementioned target. The feature of the novel self-developed SBOADM is that targeted optical signals can always be dropped down and the added upstream signal can be sent out along the reverse transmission direction of the dropped signal whether these signals are injected in either the clockwise (CW) or counterclockwise (CCW) direction. Thus, optical signals generated from the central office (CO) can be transmitted to each SBOADM node in either CW or CCW directions of the ring structure network, and the upstream signals added by SBOADMs can be sent back to the CO in the reverse optical pathways to eliminate the impact of fiber link failure. Complicated network management and additional power are unnecessary because the overall architecture of the SBOADM comprises passive components.
[25] Park, S.B.; Lee, C.H.; Kang, S.G.; Lee, S.B. Bidirectional WDM self-healing ring network for hub/remote nodes. IEEE Photonics Technol. Lett. 2003, 15, 1657–1659.
- I suggest the authors show the BER curve results of the detected downlink or uplink signals. Such curves can give a clearer explanation of the quality and stability of the signals.
Response: Thanks to the reviewer’s suggestion. The eye diagrams were measured over a period of time. As a device, the measured clear and open eyes will be great evidences to ensure the quality and stability of the SBOADM to be placed in optical fiber transport systems or fiber sensor networks. Once if the SBOADM is employed in a transport system, the BER curve results for the detected downlink and uplink signals will be presented. An additional paragraph has been added in the third paragraph of the Experimental Results and Discussion section for better understanding.
The eye diagrams were measured over a period of time. The clear and open eyes will be great evidences to ensure the quality and stability of the SBOADM to be placed in optical fiber transport systems or fiber sensor networks. When the proposed SBOADM is placed in a ring topology network, no complicate management or power supply is required to achieve the add/drop function. The 1.2 Gbps / 10 GHz signal is used mainly to verify the transmission feasibility of the proposed architecture. The overall Bragg wavelength range of FBGs within the SBOADM can be extended by a wider–bandwidth FBG or cascading multiple FBGs. As a result, different modulation format, such as 400 Gbps PAM or OFDM signals, can also been employed in the SBOADM-based optical fiber network.
- It seems the signals in Figure 4 (a) and Figure 7 (a) have high noise level in the low frequency range. Why does this happen?
Response: As the 1.2Gbps signal is raised to 10 GHz after passing through the amplifier, a large noise level is produced in the low frequency range as the amplifier gain is too large. However, it does not affect the subsequent frequency reduction and demodulation, so the signal after demodulation is quite complete.
- Some details about the FBGs should be given and what’s the optimal reflection characteristic of the FBGs?
Response: Thanks to the reviewer’s suggestion. The experimental setup is modified for better understand and a table is added to present the system parameters in the Experimental Setup section.
The schematic of the proposed SBOADM architecture is shown in Fig. 1. This architecture uses only two types of common passive components: four 4-port OCs and two FBGs with 99% reflectivity (The parameters are shown in Table.1 in details). The red auxiliary line path (Dropped signal) in Fig.1 (a) shows that the downlink optical signal will be routed to FBG1 through OC1 (entered from port 2 and outputted from port 3) when the downlink optical signal is inputted to the I/O_P1 port of this new type SBOADM. Subsequently, the optical signal located at the FBG1 Bragg wavelength will be reflected back to OC1 (entered from port 3 and outputted from port 4) and OC2 (enter from port 1 and output from port 2) and then routed to the Add/Drop_1 port. In addition, the other downlink optical signals outside the FBG1 Bragg wavelength range will be guided through FBG1 and OC3 (enters from port 2 and outputted from port 3) and outputted to the I/O_P2 port, as indicated by the blue auxiliary line path (Passed signal) in Fig. 1 (a). Simultaneously, when the uplink optical signal enters from the Add/Drop_1 port, this signal will be guided through OC2 (enters from port 2 and outputted from port 3) and then routed to FBG2. When the uplink optical carrier wavelength is located at the FBG2 Bragg wavelength range, the uplink optical signal will be reflected back to OC2 by FBG2 and outputted by OC2 (enters from port 3 and outputted from port 4) and OC1 (enters from port 1 and outputted from port 2) to I/O_P1 port, as shown in the green auxiliary line path (Added signal) in Fig. 1 (a).
Table 1. System parameters
|
Component |
Optical Circulators (OC) |
Fibre Bragg Gratings (FBG) |
|
Data Sheet |
Insertion Loss (dB):<1.3 Isolation (dB):>35 |
Reflectivity (%):99.25 Reflection Bandwidth (GHz):25 |
|
System Dimensions (mm) |
150×90×60 (L×W×H) |
|
- What’s the influence of temperature variation in the proposed scheme? As we know the wavelengths of the FBGs are temperature sensitive.
Response: Thanks to the reviewer’s comment. An additional paragraph has been added in the end of the Experimental Results and Discussion section for better understand.
Although the FBGs in 1550nm may suffer about 10 pm/℃ Bragg wavelength shifting, the effect can be eliminated by adding an appropriate thermal control scheme or simply extended the equivalent Bragg wavelength range by cascading two or more different FBGs. For example, embedding an additional 0.4nm Bragg wavelength range will eliminate the effects of 40° C temperature drifting. Besides, if there are more than one streams need to be added/dropped, the system manager can simply replace the FBGs inside the SBOADM or cascade extra FBGs with the original ones to modify the SBOADM characteristics. Once if the reflection center wavelengths of FBGs are set apart from each other, the crosstalk or interference between different wavelength channels can be minimized.
- How about when more than one streams need to be added/dropped?
Response: Thanks to the reviewer’s comment. An additional paragraph has been added in the end of the Experimental Results and Discussion section for better understand.
Although the FBGs in 1550nm may suffer about 10 pm/℃ Bragg wavelength shifting, the effect can be eliminated by adding an appropriate thermal control scheme or simply extended the equivalent Bragg wavelength range by cascading two or more different FBGs. For example, embedding an additional 0.4nm Bragg wavelength range will eliminate the effects of 40° C temperature drifting. Besides, if there are more than one streams need to be added/dropped, the system manager can simply replace the FBGs inside the SBOADM or cascade extra FBGs with the original ones to modify the SBOADM characteristics. Once if the reflection center wavelengths of FBGs are set apart from each other, the crosstalk or interference between different wavelength channels can be minimized.
Thanks for reviewer’s useful comment and suggestion.

Reviewer 2 Report
The authors have investigated a single-line bidirectional optical add/drop multiplexer (SBOADM) to support the stability and reliability of ring topology optical fiber transport systems. The manuscript is interesting and useful for the design and application of optical fiber networks. The paper is acceptable to be published in Sensors, provided the following issue can be addressed
- Some abbreviations should be clarified when they appear for the first time.
- Check the format and make sure that it is consistent with Sensors template.
- The language of this manuscript can be improved a bit with careful check. The language may need to be changed to US style instead of UK style.
- Add a table to describe the physical parameters of the experimental setup and system dimensions.
- Discuss if the crosstalk between different wavelength channels will affect the performance of the transmission experiment.
- Add some discussion on the impact from modulation format in the considered optical fiber network.
Author Response
Manuscript ID: sensors-1137692
Title: Single-Line Bidirectional Optical Add/Drop Multiplexer for Ring Topology Optical Fibre Networks
Author: Chung-Yi Li, Ching-Hung Chang *, Zih-Guei Lin
The Reply to Reviewer 2
We thank the reviewer for the comments and valuable suggestions. The manuscript has been revised accordingly. All modifications are indicated with “underline” for reviewing convenience.
Comments to the Author
The authors have investigated a single-line bidirectional optical add/drop multiplexer (SBOADM) to support the stability and reliability of ring topology optical fiber transport systems. The manuscript is interesting and useful for the design and application of optical fiber networks. The paper is acceptable to be published in Sensors, provided the following issue can be addressed
- Some abbreviations should be clarified when they appear for the first time.
Response: Thanks to the reviewer’s comment. The manuscript has been amended in the content.
- Check the format and make sure that it is consistent with Sensors template.
Response: Thanks to the reviewer’s comment. The manuscript has been amended in the content.
- The language of this manuscript can be improved a bit with careful check. The language may need to be changed to US style instead of UK style.
Response: Thanks to the reviewer’s comment. The manuscript has been amended in the content.
- Add a table to describe the physical parameters of the experimental setup and system dimensions.
Response: Thanks to the reviewer’s suggestion. The experimental setup is modified for better understand and a table is added to present the system parameters in the Experimental Setup section.
The schematic of the proposed SBOADM architecture is shown in Fig. 1. This architecture uses only two types of common passive components: four 4-port OCs and two FBGs with 99% reflectivity (The parameters are shown in Table.1 in details). The red auxiliary line path (Dropped signal) in Fig.1 (a) shows that the downlink optical signal will be routed to FBG1 through OC1 (entered from port 2 and outputted from port 3) when the downlink optical signal is inputted to the I/O_P1 port of this new type SBOADM. Subsequently, the optical signal located at the FBG1 Bragg wavelength will be reflected back to OC1 (entered from port 3 and outputted from port 4) and OC2 (enter from port 1 and output from port 2) and then routed to the Add/Drop_1 port. In addition, the other downlink optical signals outside the FBG1 Bragg wavelength range will be guided through FBG1 and OC3 (enters from port 2 and outputted from port 3) and outputted to the I/O_P2 port, as indicated by the blue auxiliary line path (Passed signal) in Fig. 1 (a). Simultaneously, when the uplink optical signal enters from the Add/Drop_1 port, this signal will be guided through OC2 (enters from port 2 and outputted from port 3) and then routed to FBG2. When the uplink optical carrier wavelength is located at the FBG2 Bragg wavelength range, the uplink optical signal will be reflected back to OC2 by FBG2 and outputted by OC2 (enters from port 3 and outputted from port 4) and OC1 (enters from port 1 and outputted from port 2) to I/O_P1 port, as shown in the green auxiliary line path (Added signal) in Fig. 1 (a).
Table 1. System parameters
|
Component |
Optical Circulators (OC) |
Fibre Bragg Gratings (FBG) |
|
Data Sheet |
Insertion Loss (dB):<1.3 Isolation (dB):>35 |
Reflectivity (%):99.25 Reflection Bandwidth (GHz):25 |
|
System Dimensions (mm) |
150×90×60 (L×W×H) |
|
- Discuss if the crosstalk between different wavelength channels will affect the performance of the transmission experiment.
Response: Thanks to the reviewer’s comment. An additional paragraph has been added in the end of the Experimental Results and Discussion section for better understand.
Besides, if there are more than one streams need to be added/dropped, the system manager can simply replace the FBGs inside the SBOADM or cascade extra FBGs with the original ones to modify the SBOADM characteristics. Once if the reflection center wavelengths of FBGs are set apart from each other, the crosstalk or interference between different wavelength channels can be minimized.
- Add some discussion on the impact from modulation format in the considered optical fiber network.
Response: Thanks to the reviewer’s comment. An additional paragraph has been added in the third paragraph of the Experimental Results and Discussion section for better understanding.
The 1.2 Gbps / 10 GHz signal is used mainly to verify the transmission feasibility of the proposed architecture. The overall Bragg wavelength range of FBGs within the SBOADM can be extended by a wider–bandwidth FBG or cascading multiple FBGs. As a result, different modulation format, such as 400 Gbps PAM or OFDM signals, can also been employed in the SBOADM-based optical fiber network.
Thanks for reviewer’s useful comment and suggestion.

Reviewer 3 Report
Single-Line Bidirectional Optical Add/Drop Multiplexer for Ring Topology Optical Fibre Networks
- The abstract seems to be long, and they should consider summarizing parts of it, so it flows better.
- They should consider having a baseline for their measurement system and give experimental parameters and details are lacking.
- Also, how does their multiplexer performance compare with existing
- The paper does not present any model to predict or understand the observed behavior.
Author Response
Manuscript ID: sensors-1137692
Title: Single-Line Bidirectional Optical Add/Drop Multiplexer for Ring Topology Optical Fibre Networks
Author: Chung-Yi Li, Ching-Hung Chang *, Zih-Guei Lin
The Reply to Reviewer 3
We thank the reviewer for the comments and valuable suggestions. The manuscript has been revised accordingly. All modifications are indicated with “underline” for reviewing convenience.
Comments to the Author
Single-Line Bidirectional Optical Add/Drop Multiplexer for Ring Topology Optical Fibre Networks
- The abstract seems to be long, and they should consider summarizing parts of it, so it flows better.
Response: Thanks to the reviewer’s suggestion. The abstract has been modified.
A new type of passive single-line bidirectional optical add/drop multiplexer (SBOADM) is proposed and experimentally demonstrated. When the proposed SBOADM is placed as a node of a ring topology optical fiber network, the special routing function of the SBOADM can always drop down the desired downstream signals whether the signals are injected into the SBOADM in either the clockwise (CW) or counterclockwise (CCW) direction and can upload and send back the upstream signals via the reversed optical pathway of the downstream signals. Once fiber link failure occurs in the optical network, the blocked network connections can be recovered immediately by sending out the downstream signals in both the CW and CCW directions of the fiber ring. As an all–passive devices, the SBOADM needs no power supply or complicated network management to achieve the bidirectional function. Thus, the proposed device is an optimal solution to enhance the stability and reliability of rapidly developed optical fiber networks.
- They should consider having a baseline for their measurement system and give experimental parameters and details are lacking.
Response: Thanks to the reviewer’s suggestion. The experimental setup has been modified for better understand and a table is added to present the system parameters in the Experimental Setup section.
The schematic of the proposed SBOADM architecture is shown in Fig. 1. This architecture uses only two types of common passive components: four 4-port OCs and two FBGs with 99% reflectivity (The parameters are shown in Table.1 in details). The red auxiliary line path (Dropped signal) in Fig.1 (a) shows that the downlink optical signal will be routed to FBG1 through OC1 (entered from port 2 and outputted from port 3) when the downlink optical signal is inputted to the I/O_P1 port of this new type SBOADM. Subsequently, the optical signal located at the FBG1 Bragg wavelength will be reflected back to OC1 (entered from port 3 and outputted from port 4) and OC2 (enter from port 1 and output from port 2) and then routed to the Add/Drop_1 port. In addition, the other downlink optical signals outside the FBG1 Bragg wavelength range will be guided through FBG1 and OC3 (enters from port 2 and outputted from port 3) and outputted to the I/O_P2 port, as indicated by the blue auxiliary line path (Passed signal) in Fig. 1 (a). Simultaneously, when the uplink optical signal enters from the Add/Drop_1 port, this signal will be guided through OC2 (enters from port 2 and outputted from port 3) and then routed to FBG2. When the uplink optical carrier wavelength is located at the FBG2 Bragg wavelength range, the uplink optical signal will be reflected back to OC2 by FBG2 and outputted by OC2 (enters from port 3 and outputted from port 4) and OC1 (enters from port 1 and outputted from port 2) to I/O_P1 port, as shown in the green auxiliary line path (Added signal) in Fig. 1 (a).
Table 1. System parameters
|
Component |
Optical Circulators (OC) |
Fibre Bragg Gratings (FBG) |
|
Data Sheet |
Insertion Loss (dB):<1.3 Isolation (dB):>35 |
Reflectivity (%):99.25 Reflection Bandwidth (GHz):25 |
|
System Dimensions (mm) |
150×90×60 (L×W×H) |
|
- Also, how does their multiplexer performance compare with existing
Response: The traditional OADM-based optical transmission systems only supports a single transmission direction, while the proposed SBOADM can support bidirectional transmission by utilizing all-passive devices. The Introduction section has been modified for better understanding.
The traditional fiber-optic network architecture is mainly designed in tree topology because of cost consideration. However, such a topology has poor performance in the reliability and the flexibility of adding or removing network nodes. When using the current network architecture whilst considering the cost of equipment build-up, some researchers have suggested that a mixed tree and ring network architecture may be a reasonable solution [8]. In a ring-shaped architecture, considering scalability and system reliability is necessary; thus, the optical add/drop multiplexer (OADM) is a good choice [9-11]. However, OADM-based optical transport systems only support the single transmission direction, and the traditional bidirectional OADM-based optical network requires additional deployment of spare optical fiber loops in the transport system to ensure network connection during fiber link failure [12-14]. Such devices cannot maximize the potential of utilizing bidirectional transmissions to overcome network traffic disconnections during fiber link failure in a ring topology network. Therefore, other researchers have proposed that reconfigurable OADM (ROADM), which can meet the needs of network wavelength allocation, can achieve the characteristics of capturing arbitrary wavelengths. However, building ROADM in the access network is not suitable due to its high cost and complex structure [15-16]. Several types of single-line bidirectional add/drop multiplexers (SBOADMs) [17-25], which can operate bidirectional transmissions on a single fiber loop, are later developed. The overall network construction cost can be reduced by removing the spare optical fiber loop. Nevertheless, such SBOADMs need the assistance of optical switches, optical multiplexers or de-multiplexers to achieve bidirectional function in a ring topology network. When SBOADMs are employed to deploy RoF transport systems, the overall cost and insertion loss caused by such SBOADMs are unignorably. A complicated strategy is required to manage the status of various optical switches inside such SBOADMs and reset the network route pathway dynamically during fiber link failure.
The SBOADM, as a network node placed in ring topology-based optical fiber transport systems, becomes appropriate if its architecture is further improved. A new form of SBOADM is proposed on the basis of four 4-port optical circulators (OCs) and two fiber Bragg gratings (FBGs) to achieve the aforementioned target. The feature of the novel self-developed SBOADM is that targeted optical signals can always be dropped down and the added upstream signal can be sent out along the reverse transmission direction of the dropped signal whether these signals are injected in either the clockwise (CW) or counterclockwise (CCW) direction. Thus, optical signals generated from the central office (CO) can be transmitted to each SBOADM node in either CW or CCW directions of the ring structure network, and the upstream signals added by SBOADMs can be sent back to the CO in the reverse optical pathways to eliminate the impact of fiber link failure. Complicated network management and additional power are unnecessary because the overall architecture of the SBOADM comprises passive components.
- The paper does not present any model to predict or understand the observed behavior.
Response: Thanks to the reviewer’s suggestion. The Experimental Setup section has been modified for better understanding.
The schematic of the proposed SBOADM architecture is shown in Fig. 1. This architecture uses only two types of common passive components: four 4-port OCs and two FBGs with 99% reflectivity (The parameters are shown in Table.1 in details). The red auxiliary line path (Dropped signal) in Fig.1 (a) shows that the downlink optical signal will be routed to FBG1 through OC1 (entered from port 2 and outputted from port 3) when the downlink optical signal is inputted to the I/O_P1 port of this new type SBOADM. Subsequently, the optical signal located at the FBG1 Bragg wavelength will be reflected back to OC1 (entered from port 3 and outputted from port 4) and OC2 (enter from port 1 and output from port 2) and then routed to the Add/Drop_1 port. In addition, the other downlink optical signals outside the FBG1 Bragg wavelength range will be guided through FBG1 and OC3 (enters from port 2 and outputted from port 3) and outputted to the I/O_P2 port, as indicated by the blue auxiliary line path (Passed signal) in Fig. 1 (a). Simultaneously, when the uplink optical signal enters from the Add/Drop_1 port, this signal will be guided through OC2 (enters from port 2 and outputted from port 3) and then routed to FBG2. When the uplink optical carrier wavelength is located at the FBG2 Bragg wavelength range, the uplink optical signal will be reflected back to OC2 by FBG2 and outputted by OC2 (enters from port 3 and outputted from port 4) and OC1 (enters from port 1 and outputted from port 2) to I/O_P1 port, as shown in the green auxiliary line path (Added signal) in Fig. 1 (a).
Figure 1. Schematic of the proposed SBOADM in the following cases: (a) optical signals inserted from the left-hand side and (b) optical signals inserted from the right-hand side.
Table 1. System parameters
|
Component |
Optical Circulators (OC) |
Fibre Bragg Gratings (FBG) |
|
Data Sheet |
Insertion Loss (dB):<1.3 Isolation (dB):>35 |
Reflectivity (%):99.25 Reflection Bandwidth (GHz):25 |
|
System Dimensions (mm) |
150×90×60 (L×W×H) |
|
For optical signals inputted from the I/O_P2 port of the proposed SBOADM, non-target optical signals, as shown in the blue auxiliary line path (Passed signal) in Fig. 1 (b), will pass through OC3 (enters from port 3 and outputted from port 4), OC4 (enters from port 1 and outputted from port 2), FBG2, OC2 (enters from port 3 and outputted from port 4) and OC1 (enters from port 1 and outputted from port 2) before being routed to the I/O_P1 port. Optical signals at target wavelengths inserted from the I/O_P2 port also pass through OC3 (enters from port 3 and outputted from port 4) and OC4 (enters from port 1 and outputted from port 2) but are reflected back to OC4 (enters from port 2 and outputted from port 3) by FBG2 and outputted to the Add/Drop_2 port, as shown in the red auxiliary line path (Dropped signal) of Fig. 1 (b). When the optical signal is inputted through the Add/Drop_2 port, this signal will be guided through OC4 (enters from port 3 and outputted from port 4) and OC3 (enters from port 1 and outputted from port 2) and then reflected back to OC3 (enters from port 2 and outputted from port 3) via FBG1 before being outputted to the I/O_P2 port, as shown in the green auxiliary line path (Added signal) in Fig. 1 (b).
Fig. 2 (a) shows that five downstream optical carriers are fed into the new type SBOADM via I/O_P1 and I/O_P2 ports to evaluate the drop/add function of the proposal, and one upstream optical carrier is added in the SBOADM via Add/Drop_1 and Add/Drop_2 ports. Each downstream optical carrier has the same power of 0.06 dBm but different wavelengths whose center wavelengths are 1544.540, 1546.540, 1548.540, 1550.540 and 1552.540 nm. Each optical carrier is modulated with 1.2 Gbps/10 GHz radio-frequency (RF) signal via Mach–Zender modulator. The third downstream optical carrier is also reused as an upstream optical carrier (as shown in Fig. 2 (b)) to simplify the experiment, and the Bragg wavelengths of the FBG1 and FBG2 are set at the same range of 1548.540 nm.
Thanks for reviewer’s useful comment and suggestion.

Round 2
Reviewer 1 Report
In their revised version of the manuscript the authors have satisfactorily addressed the concerns by myself. The paper is now ready for publication in Sensors